# SiC-IrSi_3_ for High Oxidation Resistance

**DOI:** 10.3390/ma13010098

**Published:** 2019-12-24

**Authors:** Antonio Camarano, Donatella Giuranno, Javier Narciso

**Affiliations:** 1Instituto Universitario de Materiales de Alicante (IUMA), Universidad de Alicante, Apdo. 99, 03080 Alicante, Spain; antoniodaniel.camarano@ua.es; 2Institute of Condensed Matter Chemistry and Energy Technologies (ICMATE), National Research Council of Italy (CNR), Via De Marini 6, 16149 Genoa, Italy; 3Departamento de Química Inorgánica, Universidad de Alicante, Apdo. 99, 03080 Alicante, Spain

**Keywords:** SiC composites, reactive infiltration, silicides

## Abstract

SiC is a material with excellent mechanical and thermal properties but with a high production cost. Obtaining SiC by reactive infiltration is an attractive method with a much lower cost than the traditional sintering process. However, the reactive infiltration process presents a serious problem, which is the high residual silicon content, which decreases its applicability. The replacement of silicon with silicides is a widely used alternative. The present investigation shows the good mechanical properties of the SiC-IrSi_3_ composite material obtained by reactive infiltration of SiC-C preforms with Ir–Si alloys. The thermomechanical analysis shows a high compatibility of silicide with SiC. The presence of the silicide shows a substantial improvement against the oxidation of the SiC-Si composites.

## 1. Introduction

Currently, SiC is perhaps the ceramic material with the widest applications, ranging from catalyst support [1] to metal matrix composite materials [2], due to its excellent thermal and mechanical properties [3,4]. Its use in extreme conditions (including high temperature, high heat flux, and neutron radiation) has made SiC the ideal candidate for these applications. These excellent properties are derived from the great strength of its covalent bond [5,6]. In addition, the covalent bond causes low self-diffusion, so the densification of this ceramic material is extremely complex [5,6]. Usually, temperatures above 2000 °C and pressures above 20 MPa are necessary. It is also necessary to add a certain amount of B_2_O_3_ and/or graphite for optimum sintering [7,8]. A modification of this technology is the nanoinfiltration and transient eutectic (NITE) specially designed for use in nuclear technology [8,9]. In both classical and modified technology, the cost of the material is extremely high. Although sintering routes such as spark plasma [10] are still being sought, their cost-to-performance ratio is still too high. Nevertheless, 50 years ago, a new material called reaction-bonded SiC (RBSC) appeared. The advantage of this material was twofold: the conditions of its synthesis were less severe (1600 °C and ambient pressure), and complex shapes could be made. The idea was quite simple: to infiltrate porous preforms of a mixture of α-SiC and graphite with liquid silicon, the liquid silicon reacts with the carbon to give ß-SiC, and the α-SiC acts as a seed for its growth, forming a dense network of SiC [11,12]. Subsequently, it could be demonstrated that the presence of SiC was not necessary to obtain a SiC with excellent properties [13,14,15]. The main problem for high-temperature applications (> 1000 °C) of this material is that the microstructure has unreacted Si (content greater than 10%), which causes creep problems [16]. In the last 25 years, the idea of transforming silicon into silicides has been worked upon, such as with the SiC-MoSi_2_ system [17,18]. This composite material is used in industry nowadays. Another series of composite materials has also been developed by reactive infiltration of Si-M alloys (M = Fe, Co, Ni, Al) [19,20,21,22,23], some of them with excellent properties for advanced armor systems [24]. On the other hand, the use of iridium as an oxidation protector in extreme conditions is well-established, such as in the leading edge of the space shuttle [25]. Ir is practically impermeable by oxygen at temperatures below 2000 °C, as is the case with Ir-Si alloys [26,27]. Therefore, it is logical to think that a SiC-IrSi_X_ composite material will be a material with high resistance to oxidation. Previous studies by the authors have analyzed the compatibility of SiC with Ir [28] and the process of reactive infiltration of Si-Ir alloys in SiC-C porous preforms [29], where the infiltration process has been optimized, and SiC-IrSi_3_ composite has been manufactured.

This research focuses on the manufacture and characterization of the SiC-IrSi_3_ composite material and its advantages over a standard RBSC material. An exhaustive analysis of the microstructure of the materials obtained, their mechanical and thermomechanical properties, and their resistance to oxidation has been carried out.

## 2. Materials and Methods

A detailed description of the materials and their characterization has been presented in previous articles [30,31]. In this section, we will only detail the most relevant data for this research.

### 2.1. Materials

The SiC porous-C preforms used in this work were kindly provided by the company Petroceramic (Bergamo, Italy). These preforms consist of a bimodal mixture of α-SiC particles, with a size of 36.5 and 4.5 μm respectively, joined by a hard carbon matrix. The samples were supplied by the company in the form of prisms with dimensions of 25 × 25 × 5 mm, from which substrates of 5 × 5 × 5 mm were cut off for the manufacturing of SiC-Si-IrSi_3_ composite materials. Prior to their use, the SiC-C preforms were thermally treated at 1500 °C to avoid the appearance of volatile compounds during the experimental process (see Table 1 of reference [29]). 

Si-62 wt% Ir alloys of eutectic composition [32] were prepared by mixing iridium and high-purity silicon. The iridium wire of 0.25 mm diameter used was supplied by Goodfellow (Cambridge, UK) with a purity of 99.9% (REF: 980-032-570). The silicon was obtained from a silicon wafer (Wafer) of high purity (99.9999%). The alloys were prepared in preformed drops of 400.0 mg for manufacturing composite materials by reactive infiltration. The method selected for the synthesis of alloys was the arc melting process under vacuum atmosphere (0.001 MPa). The equipment used was a furnace brand Edmund Bühler GmbH (Bodelshausen, Germany) model Compact Arc Melter MAM-1. After each fusion process, the alloy drops were ultrasonically cleaned with ethanol for 5 min and air-dried to remove possible contaminants. The fusion and cleaning processes were repeated for 4 times to ensure the homogeneity of the alloy composition. After that, prior to its use, an alloy of each batch was characterized to verify its composition and microstructure. To observe the microstructure by SEM microscopy, the alloy was embedded in epoxy resin and polished with standard. Figure 1 shows a representative SEM image of the polished cross section of the eutectic alloy.

### 2.2. Experimental Procedure

The experiments series were carried out in an induction horizontal furnace heated by an 800 kHz high frequency generator coupled to a graphite susceptor. The series of experiments were specially monitored by a high-resolution camera connected to a computer with an image analysis system (ASTRA) [33,34] used for the automatic acquisition of geometric variables in real time. During all the experiments, the temperature was monitored with a pyrometer, which was previously calibrated by measuring the melting temperature of Cu and Ni, both with a high purity (99.99%).

Experiments always started by placing the preform and the metal on a specially designed graphite crucible and positioning the SiC-C porous preform at the bottom and the silicon or Si-62 wt% Ir alloy on top. Then, the whole set was introduced into the central part of the furnace and aligned with the measurement optical system. Once aligned, the furnace was closed and the chamber was degassed under high vacuum (10^−5^ Pa) for two hours in order to clean and eliminate any contaminant in the system. Afterward, the temperature was raised to the selected value, using a heating speed of 2 °C/s, always under a vacuum atmosphere (10^−5^ Pa). For the synthesis of SiC-Si and SiC-IrSi_3_ composite materials by reactive infiltration, temperatures of 1450 °C and 1350 °C, respectively, were used. The materials were synthesized by triplicate using SiC-C preforms with dimensions of 5 × 5 × 5 mm and metal drops preformed with a weight of 400 mg. In order to preserve the infiltration microstructure and study the thermal compatibility in composite materials, we decided to perform a rapid cooling (higher than 1 °C/s) in all cases. Once the experiments were finished, the samples were removed from the furnace and embedded in polymeric resin inside a vacuum chamber. This procedure made it possible to completely fill the remaining porosity. The resin samples were cross-sectioned and polished using standard metallographic techniques (diamond cloths with grain size P600, P1200, 5 μm, 1 μm, and 0.25 μm). Composition and microstructure were checked using scanning electron microscopy with X-ray microanalysis (SEM-EDS), both in the backscattered electron (BSE) and secondary electron (SE) mode. In addition, the crystalline phases were recognized by X-ray diffraction (XRD).

In relation to thermomechanical characterization of SiC-Si and SiC-Si-IrSi_3_ composite materials, the coefficient of thermal expansion (CTE), Vickers microhardness (HVN), and a guide value of fracture toughness were evaluated. CTE was obtained using a TMA 2940 thermomechanical analyzer, (TA Instruments, New Castle, DL, USA). The thermal response curves were obtained by applying a force of 0.05 N to the samples with dimensions of 5 × 5 × 5 mm under nitrogen atmosphere in the temperature range of 25–900 °C. Samples were submitted to 3 heating and cooling cycles to eliminate any residual stress that might have been developed during the processing of the composite material. For Vickers microhardness measurements, a microindentator (Buehler, Micromet 2100, Chicago, IL, USA) with a load of 300 g for silicon and 1000 g for alloys and composite materials was used. In all cases, a retention time of 15 s was used. Vickers microhardness value was calculated based on the applied load and the average size of the printing diagonals caused by a diamond indenter (regular-shaped pyramid). This test also provides an indirect measuring of the fracture toughness of the material (*K_C_*), called fracture toughness to the indentation (*K_C_*). This property, with MPa·m^1/2^ units, is calculated by the following semi-empirical equation proposed by Anstis [35]
(1)$$K_{IC}=\chi {\left(\frac{E}{H}\right)}^{1/2}\frac{P}{a^{3/2}}$$
where *P* is the applied load (MN), *E* is the Young modulus (MPa), *H* is the Vickers hardness (MPa), *a* is the radial length of the fracture measured from the center of the trace (m) and *χ* is a constant of empirically determined calibration, determined as 0.016 ± 0.004. This formula, considered as a direct measure of fracture toughness, has been strongly criticized by several authors since it has a certain degree of error in its formulation [35] (errors between 25% and 30%). Additionally, it incurs data-interpretation errors since the fracture measurement depends on the microscopes used and the perception of the operator. For this reason, it is not accepted as valid to faithfully calculate fracture toughness. However, it can be used to compare similar materials [36], as is the case with this investigation.

Oxidation behavior of SiC-Si and SiC-Si-IrSi_3_ composite materials was investigated by thermogravimetric analysis. For these tests, a model device (TGA/SDTA851e/SF/1100, Mettler Toledo, Columbus, OH, USA) was used. Two types of analysis were carried out. In the first one, an oxidizing atmosphere (N_2_/O_2_ = 4:1) and temperatures between 25 °C and 1200 °C were used with a heating ramp of 5 °C/min in order to observe the iridium effect on the oxidation resistance of the composite material. In the second analysis, a series of experiments were performed under isothermal conditions for 5 h at three different temperatures, 1000, 1100, and 1200 °C. Just to observe the oxidation on the isothermal plane, the temperature was elevated under a nonoxidizing atmosphere (N_2_) with a heating ramp of 10 °C/min, and a moment before reaching the isothermal condition, it was changed to an oxidizing atmosphere (N_2_/O_2_ = 4:1). Analyses were carried out with gas flow of 100 cm^3^/min, using an alumina crucible. This second analysis allowed us to estimate the activation energy of the oxidation process of SiC-Si-IrSi_3_ composite materials.

## 3. Results

In the previous work, the different microstructures obtained were commented on [29]. However, the most important aspects of the composite material will be highlighted. Figure 2 shows the microstructure of the composite material, where a homogenous distribution of both the SiC and the eutectic alloy is clearly seen, with the porosity less than 3%. Through a detailed analysis of the microstructure, it was found that there was a new SiC with a particle size between 1 and 3 micrometers, which corresponds to Type II SiC. Through DRX following the procedure developed by Caccia et al. [30], it was determined that approximately 3% of the SiC was the beta phase corresponding to the new SiC found and that it fully agrees with the expected SiC, taking into account the amount of C of the original preform.

### Thermomechanical Properties of SiC-Si and SiC-Si-IrSi_3_ Materials

In this section, the hardness of precursor materials and synthesized materials was evaluated using the Vickers microindentation test. Figure 3 shows, from left to right, optical micrographs of the impressions left in pure silicon, in Si-62 wt% Ir alloy, and in SiC-Si and SiC-Si-IrSi_3_ composite materials. Additionally, microhardness values and indentation fracture toughness (*K_C_*), calculated according to the tests carried out, are shown in Figure 4.

In the first place, it is emphasized that Si-62 wt% Ir alloy has a higher hardness than pure silicon. Previous studies on iridium silicides carried out by Sha [37] indicate that values obtained in this investigation are in the same order of magnitude (for example, a hardness of 8.63 GPa for Ir_3_Si and 7.85 GPa for IrSi). Additionally, it is mentioned that this series of silicides presents brittle properties, like Mo [38] and Nb [39] silicides. The increase in the material hardness found in the interparticular zone not only gives the SiC-Si-IrSi_3_ composite a greater global hardness but also adds a lower fracture toughness, estimated by indentation fracture toughness (*K_C_*). This type of silicides, like Mo and Nb silicides, goes through a structural transition when increasing temperature, which in general is accompanied by an improvement in fracture toughness. For this reason, it is thought that at high temperature, this material would present an increase in ductility.

According to observations of the microstructure, it can be seen that there is an adequate thermal compatibility between all materials. Despite these observations, it is convenient to study this variable in depth, since it is an essential property in the behavior of materials subjected to high temperature cycles. Focusing on SiC-Si and SiC composite materials produced by reactive infiltration, several investigations have shown that they turn out to be thermally compatible combinations [40,41]. However, the use of silicon-metal alloys is often accompanied by significant thermal incompatibilities. This disparity between materials can often cause damage to microstructure during synthesis and creates a significant decrease in thermomechanical properties in all cases. As an example of this uneven behavior between phases, we have the production of SiC-iron silicide materials [20]. The great difference between thermal expansion coefficients (CTE) of iron silicides and SiC generates structural damage, making its use in thermal cycling unfeasible. For this reason, it is of vital importance to evaluate this characteristic rigorously.

Figure 5 shows the comparison of CTE evolution with the temperature of the precursor elements of synthesized composite materials. Here, it can be seen that CTE values between Si and SiC are very close, thus explaining its great thermal compatibility. Additionally, iridium references and experimental measurements of Si-62 wt% Ir alloy indicate that CTEs are of the same order of magnitude as those of SiC and Si; therefore, an excellent thermal compatibility between the alloy (matrix) and reinforcement (SiC) is expected.

The final analysis of CTE of the synthesized composite materials is shown in Figure 6. As predicted, there is a great thermal compatibility between materials, and CTE shows low values, which is compatible with high temperature structural applications. This characteristic indicates a good cohesion of the phases, both in high and low temperatures.

In this section, a study on the behavior of SiC-Si and SiC-Si-IrSi_3_ materials against oxidation was carried out. To that end, weight variation of the samples was recorded with temperature in oxidizing atmosphere by means of the TGA technique. Temperatures between 25 and 1200 °C were evaluated, with a heating ramp of 5 °C/min and an oxidizing atmosphere with an N_2_/O_2_ ratio of 4:1. It can be observed that both materials suffered from an increase in their specific weight, showing a process of passive oxidation on the surface; that is to say, the oxides formed tended to remain linked on the surface and did not evaporate. Firstly, it should be noted that although the SiC-Si system showed a homogeneous behavior, the system with silicides was completely different: at temperatures below 800 °C, it remained unchanged, which indicates that silicon is a great protector against oxidation at intermediate temperatures. However, at higher temperature, a significant increase in oxidation was observed as expected because the process of diffusion of Si in silicide was activated. This fact has been proven in a series of silicides coating silicon, where it is clear that there are two groups, including titanium, where oxygen diffusion prevails over silicon because both metal oxide and silicon oxide are formed; and others, such as Ir silicides, where silicon diffusion predominates because the formation of metal oxide is not favored.

In order to analyze this phenomenon, the study on SiC-Si-IrSi_3_ materials oxidation under isothermal conditions at temperatures above 1000 °C was carried out, where silicide also undergoes an oxidation process. This study consisted of heating the samples under an inert atmosphere up to a selected temperature, changing to an oxidizing atmosphere, and maintaining the sample in an isothermal condition for a previously defined time. In this case, an oxidizing atmosphere with an N_2_/O_2_ ratio of 4:1, an isothermal time of 5 h, and three temperatures (1000, 1100 and 1200 °C) were used. (It is important to point out that the experiments carried out with the SiC-Si composite at 1200 °C show a mass increase of 0.4 instead of 0.16 mg/m^2^, which once again shows the improvement in oxidation resistance due to the presence of iridium silicide.) Figure 7 shows the evolution results for the specific weight over time for the three selected conditions. Here, it can be seen that oxidation speeds increase steadily with temperature and that in all cases they follow a parabolic trajectory over time. It is important to highlight that oxidation speed in the Si-SiC system is several times higher than in temperatures of 1200 °C.

This series of graphs allows the observation of the general oxidation trend with temperature and also provides important information about the oxidation mechanism. Figure 8 shows an Arrhenius graph where the logarithm of speed of specific weight change is represented according to the inverse of temperature. This value turns out to be of great importance since it allows us to identify quantitatively the mechanism that governs the superficial oxidation process. According to calculations made from Figure 8, the activation energy value of SiC-IrSi_3_ materials amounts to E_Oxid_ = 88 kJ/mol. This activation energy turns out to be of the same order of magnitude as oxidation processes by diffusion [42] and nearly half the value of the activation energy of oxidation of pure silicon in dry oxygen, E_Oxid_ = 120 kJ/mol [43] (minimum value taken by SiC-Si composite materials). These differences in activation energies show that oxide formation on the surface of composite materials with iridium is energetically more unfavorable than in materials with pure silicon. This low activation energy is determined by the reaction mechanism proposed by d’Heurle [44], where silicide oxidation does not occur, and what happens is the migration of silicon to the surface and its subsequent oxidation. Under these conditions, the diffusion of oxygen does not occur as usual.

## 4. Conclusions

In this work, the viability of the production of SiC-IrSi_3_ by infiltration of SiC-C porous preforms using eutectic Si-62 wt% Ir alloys was investigated. The microstructure and mechanical properties of the different composites were measured and correlated. From a comparison between both materials, it was observed that the addition of iridium not only causes an increase in hardness but also generates greater brittleness. The thermomechanical tests showed that there is an excellent thermal compatibility between the matrix and reinforcement in all materials and that there is no evidence of phase uncoupling due to thermal expansion mismatches. Furthermore, we verified that the addition of iridium does not create any kind of incompatibility due to the close proximity of CTE of Si-62 wt% Ir alloy with the CTE of SiC and Si. On the other hand, oxidation studies showed promising results in SiC-IrSi_3_ materials.

In conclusion, SiC-IrSi_3_ composite materials appear as ideal candidates for structural applications at high temperature due to their excellent thermomechanical properties and promising oxidation resistance.

## Figures and Tables

**Figure 1 materials-13-00098-f001:**
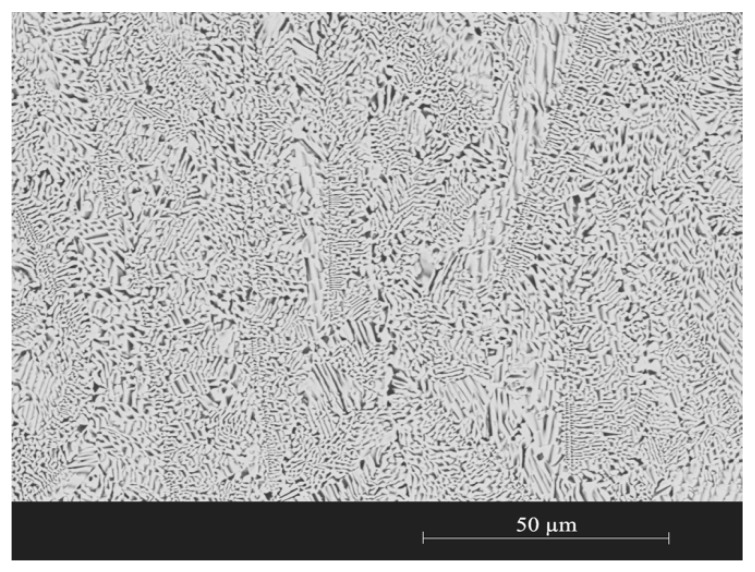
Secondary electron micrographs of the representative microstructure of eutectic alloy Si-62 wt% Ir.

**Figure 2 materials-13-00098-f002:**
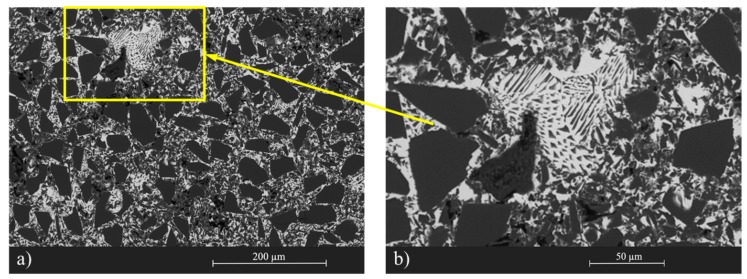
(**a**) Microstructure of the SiC-Si-IrSi3 composite material obtained by SEM; (**b**) detail of the eutectic microstructure.

**Figure 3 materials-13-00098-f003:**
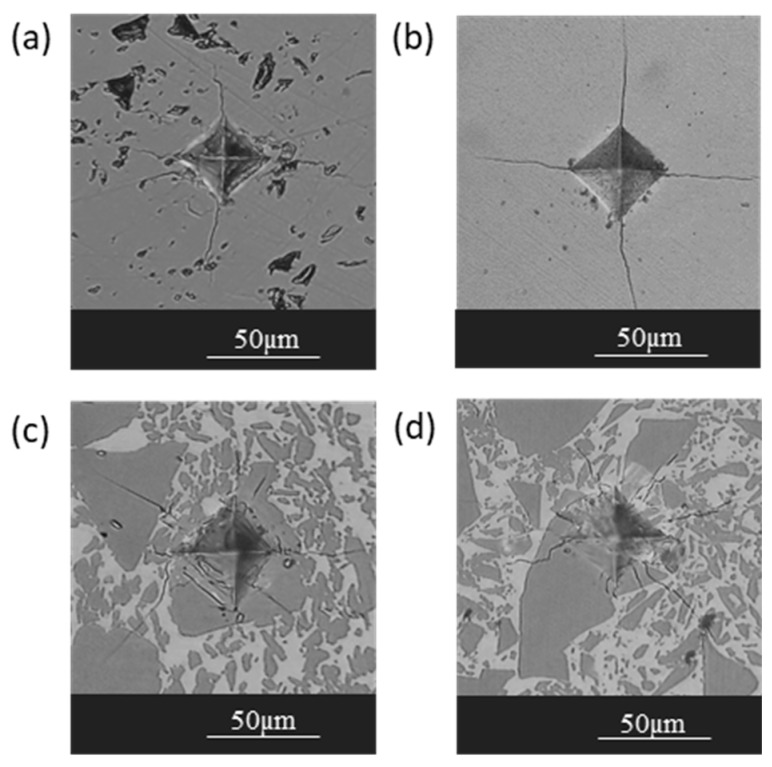
Marks left by Vickers indenter in the different materials. Detail of impressions: (**a**) pure Si, (**b**) Si-62 wt% Ir alloy, (**c**) SiC-Si composite material, and (**d**) SiC-Si-IrSi_3_ composite material.

**Figure 4 materials-13-00098-f004:**
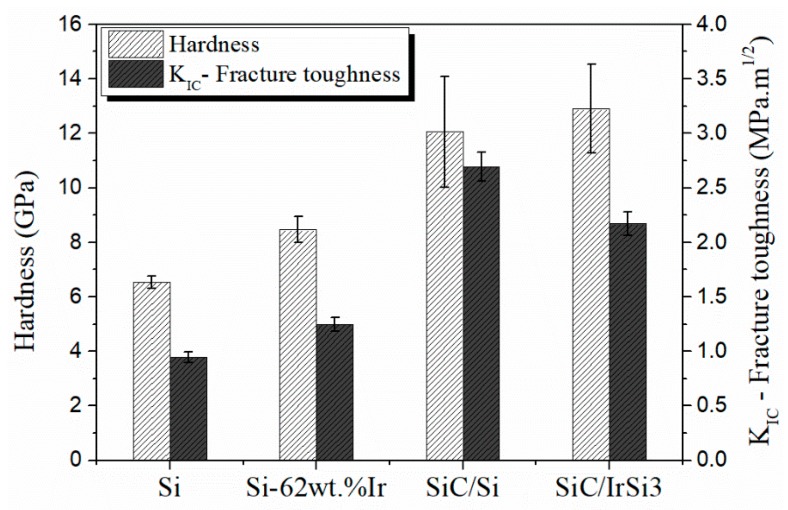
Vickers microhardness and indentation fracture toughness (KC) measure for metals (Si and Si-62 wt% Ir) and composite materials (Si-Si and SiC-IrSi_3_). Bars indicate the average value and the segments indicate the standard deviation. Five indentations were made for each sample.

**Figure 5 materials-13-00098-f005:**
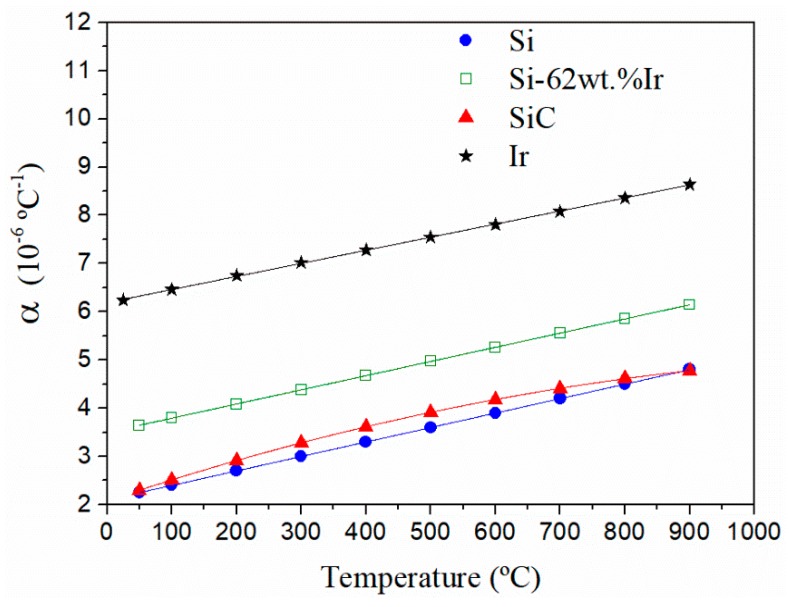
Evolution of thermal expansion coefficients, from 25 °C to 900 °C of pure silicon and Si-62 wt% Ir alloy experimentally measured, and SiC [31] and Ir [30], taken from bibliographical references.

**Figure 6 materials-13-00098-f006:**
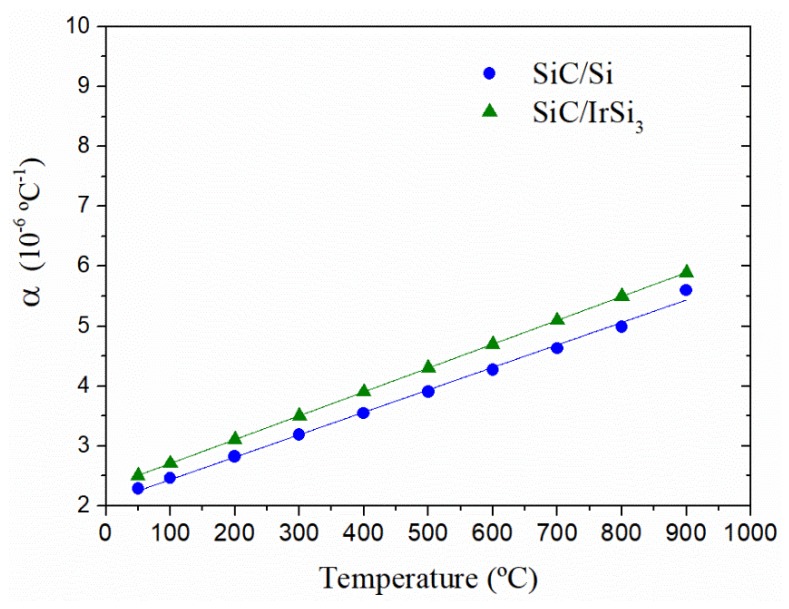
Evolution of thermal expansion coefficients, from 25 °C to 900 °C, of SiC-Si and SiC-IrSi_3_ composite materials experimentally measured.

**Figure 7 materials-13-00098-f007:**
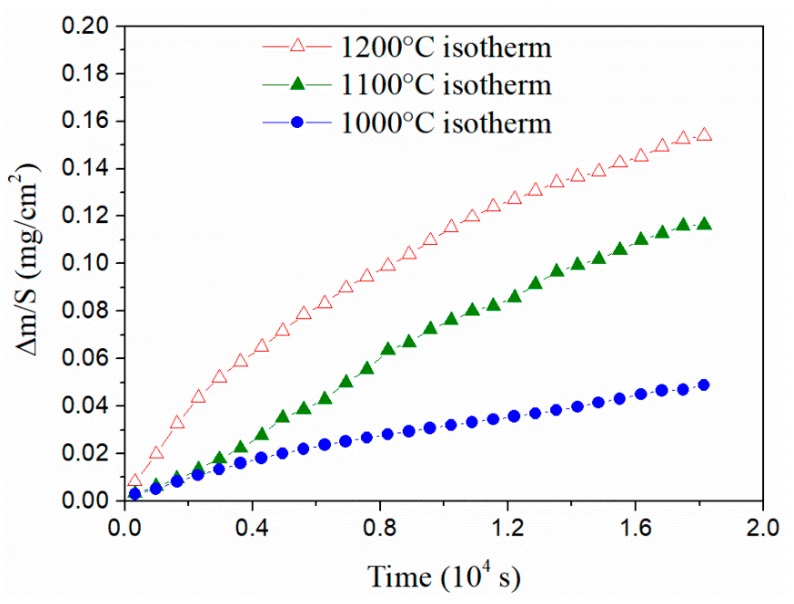
Evolution of the weight gain of the SiC-IrSi_3_ composite material obtained in an isothermal condition at different temperatures in an oxidizing atmosphere (N_2_/O_2_ ratio is 4:1).

**Figure 8 materials-13-00098-f008:**
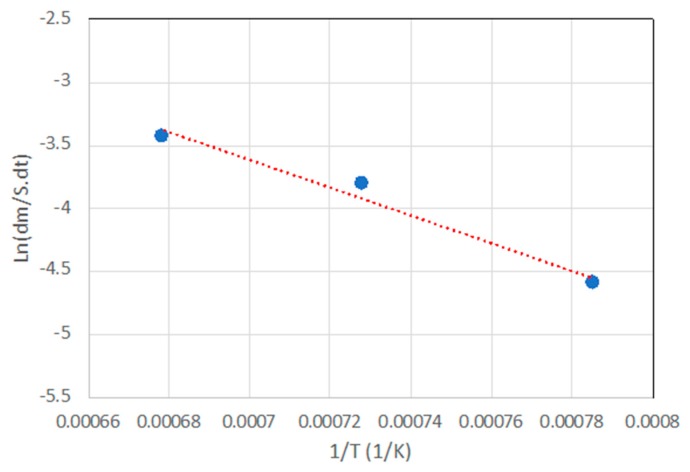
Arrhenius plot based on the data of Figure 7.

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
