# Peer review of "SiC-IrSi3 for High Oxidation Resistance"

_materials, 2019, doi:10.3390/ma13010098_

Round 1

Reviewer 1 Report

Review

SiC-IrSi3 for high oxidation resistance

by Antonio Camarano, Donatella Giuranno and Javier Narciso

The essence of this work is as follows. Heat-resistant ceramic based on SiC obtained by reactive infiltration method contains a large percentage of silicon that has not been converted to SiC. Silicon can be oxidized by atmospheric oxygen at high temperatures. As a result, this ceramic may lose some of its unique strength properties. In order to mitigate this drawback, the authors of this work proposed to convert this Si into a compound with iridium (Ir) IrSi3. As a result of such a transformation, they received a new highly resistant heat-resistant material. The article studies the basic properties of this material.

Undoubtedly, the article is very important for technical and technological applications. However, the article is purely technological, scientific novelty (research and discovery of new effects, the study of any physical, chemical and other processes) is missing. However, since this journal is a specialized journal of materials, I believe that the article can be published in its present form.

Author Response

Dear Reviewer,

Thank you very much for such valuable comments. I agree with you, it is a paper of great technological value.
The paper has been partially modified in order to improve the level of English

Reviewer 2 Report

Narciso et al. in their manuscript entitled "SiC-IrSi3 for high oxidation resistance" describe investigation on mechanical properties of the SiC/IrSi3 composite obtained by reactive infiltration of SiC/C preforms with Ir-Si alloys.

This paper is an interesting contribution to SiC composites research, includes many different techniques, and discusses results in subsequent detail.

The experimental work presented herein has been carefully designed and carried out.

Authors discuss the activation energy which was determined on the basis of thermogravimetric analysis, while this method can be used to determine the apparent activation energy. The activation energy of SiC-IrSi3 oxidation is associated with a sum of activation energies of processes running simultaneously like crystal lattice deformation, bond cleavage, mass and heat transfer and many others. Not just the simply oxidation process. This should to be discussed.

Low apparent activation energy is explained by the formation of silicon oxide. This could be confirmed by pxrd. Iridium oxide is also formed?

The manuscript has few editorial errors that can be corrected at the galley proof stage

This referee believes that presented manuscript is enough quality and is suitable for publication in Materials.

Author Response

Dear Reviewer,

Thank you very much for such valuable comments. I agree with you, in the calculation of the activity energy there are more processes involved than simple diffusion. However, we think it is the most influential factor.
We have performed PXRD of the material and under the oxidation conditions used in the present investigation we have not observed the formation of iridium oxide.

The paper has been partially modified in order to improve the level of English

Reviewer 3 Report

The manuscript has some major flaws and needs major revision.

On page 1, line 2, please remove the period at the end of the title. The authors should improve the grammar/language style/typo in the manuscript, there are some examples: On page 1, line 25, please change the “highest number of applications” with “widest applications”. On page 1, lines 27-28, “Nevertheless, its use in extreme conditions (high temperature, high heat flux, neutron radiation) has made SiC the ideal candidate for these applications.” was not clearly written. Alternative way can be “Therefore, SiC is the ideal candidate for the applications in extreme conditionings, e.g., high temperature, high heat flux, and neutron radiation.” On page 5, line 177, there is a typo and it is not consistent with line 162. Please change “Si/IrSi3 composite material” to “SiC/Si-IrSi3 composite material” On page 6, line 192, there is a typo. Please change “Si/Si” to “SiC/Si” On page 7, line 214. There is no figure showing the first type analysis to show the iridium effect on the oxidation resistance of the composite material, as mentioned on page 4 experimental section. Please add the figure to support the discussion on the behavior of SiC/Si and SiC/Si-IrSi3 materials against oxidation. In terms of the oxidation experiments, any chemical composition analysis of the materials before and after oxidation? Since there is both N2 and O2 in the ambient, both nitridation process and oxidation process can occur in such high temperatures >1000C. On page 8, the Arrhenius plot of the oxidation rate vs (1/temperature) is questionable. The 3 temperatures (1000, 1100, 1200 C) should be converted into 1273K, 1373K, 1473K first. Then, “1/temperature” should be “0.789, 0.728, 0.679 x10-3 1/K”. the author didn’t do the conversion and also used the wrong unit of 1x10-4/T”. The curves in figure 7 are not linear, the author should explain how to get the oxidation rate, e.g., the linear fitting of figure 7 from 0 to 1.8x104 s or only 1.0x104 s-1.8x104 s Please show how to get the oxidation activation energy (Eoxid) of SiC/IrSi3 and use the same method to calculate the Eoxid of SiC/Si. The authors compare with data in reference [43], if the calculation method is not consistent with other literature, then it is not an apple-to-apple comparison. The conclusion of a lower oxidation activation energy is not solid and convincing.

Author Response

Dear Reviewer,

First of all thank you for your valuable comments.

The manuscript has been revised by an expert in English grammar, and English has been improved.

Regarding your comments.

Although the temperature is extremely high (1200 ° C), I have never read that nitrides are synthesized in the presence of oxygen. It is true that at this temperature the possible formation of nitrides using ammonia has been published. Usually nitrides are synthesized at temperatures above 1400 ° C.

All samples are analyzed before and after by P-XRD. Only in the case of 1200 ° C we observe the main peak corresponding to cristobalite, but with a very low intensity. These results are compatible with the previous results carried out in our laboratory on SiC, Si3N4 and SiC-Si composites. Since it is a standard test in our laboratory. For this reason the authors think that it was not necessary to add a graph of SiC / Si oxidation. Since this can be found in various publications.

For us the important thing was to demonstrate that the presence of iridium silicide improved oxidation resistance. As stated in the text of the manuscript, up to 800 ° C is unchanged, while if there is no silicide, a small weight gain is always observed. From that temperature both materials increase their weight, but always to a greater extent in the case of SiC / Si.

The analysis that proceeds is that of the SiC-silicides system, and is the one that has been carried out. First, the shape of the curve indicates that the reaction is controlled by diffusion (). K and not ºC have been used to calculate the Ea. Although for consistency at all times we talk about ºC. The value obtained indicates that there must be an oxidation mechanism other than the conventional one. The authors have proposed a mechanism validated by other authors, although the working conditions and the method of obtaining it are different from that of this manuscript.

To propose a complete mechanism we should do many more experiments. Especially use P-XRD at high temperature. But we think that by itself would be an independent article, and that it is outside the scope of this article

Round 2

Reviewer 3 Report

The authors didn’t clearly answer my question why there is no figure showing the first type analysis, as mentioned on page 4 experimental section. The authors mentioned that “It can be observed that both materials suffer from an increase in their specific weight showing a process of passive oxidation on the surface……at temperatures below 800 °C, it remains unchanged, which indicates that silicon is a great protector against oxidation at intermediate  temperatures” There is no figure to support the discussion. Although you said that it is a standard test in your laboratory, it doesn’t mean it is not necessary to show the data. Because there is a discussion on the different behavior of two materials, without a figure, data or reference, how can you support your discussion/conclusion? The authors didn’t make any active actions to correct the problem in figure 8 and didn’t clearly answer my question. First, the curves in figure 7 are not linear, the author should explain how to get the oxidation rate in figure 8. Second, the authors replied that K and not ºC have been used to calculate the Ea. Okay, there are 3 temperatures 1000, 1100, and 1200ºC, as I said, you need to convert into 1273K, 1373K, 1473K first. Then, “1/temperature” should be 1/1273 (0.789 x10-3), 1/1373 (0.728 x10-3), 1/1473 (0.679 x10-3) 1/K”. Remember, the unit is 1/K, not 1/ ºC. The authors didn’t do the conversion, instead, the authors used 1/1000 (0.001 or 10x10-4), 1/1100 (9.09x10-4), 1/1200 (8.33x10-4), and the corresponding unit is 1/ºC. In figure 8, I can find 3 data points in the X-axis: 1.00x10-4, 0.909x10-4, and 0.833x10-4. How can you explain why there is one order of magnitude difference. The wrong data will lead to an incorrect Ea calculation. The authors should pay attention to the accuracy of your data in the figure and reviewer’s comments. The manuscript should not be accepted until the authors correct the problem in figure 8.

Author Response

First of all thank the insistence of the reviewer 3 in the error of figure 8. I have modified the figure and recalculated the activation energy. Although the value of the activation energy has changed, it is still lower than that obtained for the SiC / Si system and accepted as correct. This fact does not invalidate the explanation of the decrease in process energy, since the oxidation mechanism is different.

I have added the following statement:

(It is important to point out that the experiments carried out with the SiC/Si composite at 1200 ° C show a mass increase of 0.4 instead of 0.16 mg/m2, which once again shows the improvement in oxidation resistance due to the presence of iridium silicide)

Regarding that it is not linear. As I say in the text, the shape of the curve indicates that the reaction rate is controlled by diffusion, and the variation in mass is proportional to t (1/2). It is seen more clearly when the temperature is higher (1200 ° C), at lower temperatures it is more difficult to determine the curve because we are at the limits of microbalance detection. This indicates that the reaction rate is not constant, it decreases over time. So the reaction rate used is the initial velocity

In my humble opinion the article is correct. The message is clear, SiC composite materials can be manufactured by reactive infiltration with Si-Ir alloys. This composite has thermomechanical properties similar to SiC / Si, but the oxidation resistance increases considerably